# Genetic effects on promoter usage are highly context-specific and contribute to complex traits

Kaur Alasoo[1,2]*, Julia Rodrigues[2], John Danesh[2,3,4,5], Daniel F Freitag[2,4], Dirk S Paul[2,4], Daniel J Gaffney[2]

[1]Institute of Computer Science, University of Tartu, Tartu, Estonia; [2]Wellcome Sanger Institute, Wellcome Genome Campus, Hinxton, United Kingdom; [3]BHF Cardiovascular Epidemiology Unit, Department of Public Health and Primary Care, University of Cambridge, Cambridge, United Kingdom; [4]British Heart Foundation Centre of Excellence, Division of Cardiovascular Medicine, Addenbrooke's Hospital, Cambridge, United Kingdom; [5]National Institute for Health Research Blood and Transplant Unit (NIHR BTRU) in Donor Health and Genomics, Department of Public Health and Primary Care, University of Cambridge, Cambridge, United Kingdom

**Abstract** Genetic variants regulating RNA splicing and transcript usage have been implicated in both common and rare diseases. Although transcript usage quantitative trait loci (tuQTLs) have been mapped across multiple cell types and contexts, it is challenging to distinguish between the main molecular mechanisms controlling transcript usage: promoter choice, splicing and 3' end choice. Here, we analysed RNA-seq data from human macrophages exposed to three inflammatory and one metabolic stimulus. In addition to conventional gene-level and transcript-level analyses, we also directly quantified promoter usage, splicing and 3' end usage. We found that promoters, splicing and 3' ends were predominantly controlled by independent genetic variants enriched in distinct genomic features. Promoter usage QTLs were also 50% more likely to be context-specific than other tuQTLs and constituted 25% of the transcript-level colocalisations with complex traits. Thus, promoter usage might be an underappreciated molecular mechanism mediating complex trait associations in a context-specific manner.
DOI: https://doi.org/10.7554/eLife.41673.001

*For correspondence:
kaur.alasoo@ut.ee

## Introduction

Genome-wide association studies (GWAS) have discovered thousands of genetic loci associated with complex traits and diseases. However, identifying candidate causal genes and molecular mechanisms at these loci remains challenging. Complex trait-associated variants are enriched in regulatory elements and are therefore thought to act via regulation of gene expression levels, often in a cell type- and context-specific manner (*Alasoo et al., 2018*; *Fairfax et al., 2014*; *Kim-Hellmuth et al., 2017*). However, such variants are equally enriched among splicing quantitative trait loci (QTLs) (*Fraser and Xie, 2009*; *Li et al., 2016*) and incorporating splicing QTLs in a transcriptome-wide association study increased the number of disease-associated genes by twofold (*Li et al., 2018*). In addition to splicing, genetic variants can also alter transcript sequence by regulating promoter and 3' end usage, which we refer to collectively hereafter as transcript usage QTLs (tuQTLs). Alternative transcript start and end sites underlie most transcript differences between tissues (*Pal et al., 2011*; *Reyes and Huber, 2018*), they are dynamically regulated in response to cellular stimuli (*Alasoo et al., 2015*; *Richards et al., 2017*) and they are also frequently dysregulated in cancer (*Demircioğlu et al., 2017*; *Lee et al., 2018*). Moreover, experimental procedures designed to capture either 5' or 3'

**eLife digest** Genes contain all instructions needed to build an organism in form of DNA. Humans share around 99.5% of DNA, but it is the remaining 0.5% that contain the small genetic variations that make us unique. Subtle differences in genes can, for example, influence the color of our hair or eyes.

To build gene products, such as proteins, DNA first needs to be transcribed into RNA. Some genetic variants can affect how a gene is transcribed into an RNA molecule, for example by making it be transcribed too much or too little, which can lead to diseases. These variants can also influence where the transcription begins through a process called promoter usage. This can lead to shorter or longer RNAs, which can have different biological impacts.

With current research methods it is difficult to detect changes in the latter kind of alteration. As a result, it is harder to distinguish these from other types of changes. Now, Alasoo et al. wanted to find out what proportion of genetic variants that alter traits influence promoter usage, compared to other changes. To do so, a new computational method was developed to directly measure how genetic variants influence different parts of the RNA, such as promoters, middle sections and ends. The method was then applied to datasets of human immune cells. The experiments revealed that genetic variants often influence promoter usage. Many of the effects could only be found when cells are exposed to external stimuli, such as bacteria.

The results highlight that to discover genes responsible for human traits and disease we need to consider all the possible ways genetic differences between individuals could alter the gene products. Large published datasets could be reanalyzed using this method to identify new genes that could be implicated in human health and disease, potentially leading to new treatment options in future.

DOI: https://doi.org/10.7554/eLife.41673.002

ends of transcripts have identified disease-relevant genetic variants that regulate promoter or 3' end usage (*Garieri et al., 2017*; *Zhernakova et al., 2013*). However, well-powered RNA-seq-based tuQTL studies performed across cell types (*Battle et al., 2014*; *Chen et al., 2016*; *Lappalainen et al., 2013*; *Li et al., 2016*; *Ongen and Dermitzakis, 2015*) and conditions (*Nédélec et al., 2016*; *Ye et al., 2018*) have thus far not distinguished between promoter usage, splicing and 3' end usage. Thus, how these distinct transcriptional mechanisms contribute to complex traits and how context-specific these genetic effects are is currently unclear.

In addition to splicing analysis, RNA-seq data can also be used to quantify promoter and 3' end usage. The simplest approach would be to first quantify the expression of all annotated transcripts using one of the many quantification algorithms (benchmarked in *Teng et al., 2016*). Linear regression can then be used to identify genetic variants that are associated with the usage of each transcript of a gene (*Li et al., 2018*; *Ongen and Dermitzakis, 2015*). Comparing the associated transcripts to each other can reveal which transcriptional changes take place (*Figure 1A*). A key assumption here is that all expressed transcripts are also part of the annotation catalog. If some of the expressed transcripts are missing, then reads originating from the missing transcripts might be erroneously assigned to other transcripts that are not expressed at all (*Figure 1B*) (*Soneson et al., 2018*). This can lead to individual genetic variants being spuriously associated with multiple transcriptional changes. For example, a genetic variant regulating promoter usage might also appear to be associated with the inclusion of an internal exon (*Figure 1B*), although there are no reads originating from that exon. Importantly, this is not just a theoretical concern, because 25–35% of the exon-exon junctions observed in RNA-seq data are not present in transcript databases (*Ongen and Dermitzakis, 2015*), and up to 60% of the transcripts annotated by Ensembl (*Zerbino et al., 2018*) are truncated at the 5' or 3' end (*Figure 1—figure supplement 1*, *Figure 1—figure supplement 2*).

To overcome the issue of missing transcript annotations, recent tuQTL studies have focussed on quantifying transcription at the level of individual exons (*Fadista et al., 2014*; *Lappalainen et al., 2013*; *Odhams et al., 2017*), introns (*Odhams et al., 2017*) or exon-exon junctions (*Figure 1C*) (*Li et al., 2018*; *Odhams et al., 2017*; *Ongen and Dermitzakis, 2015*). While these approaches often discover complementary genetic associations (*Odhams et al., 2017*; *Ongen and Dermitzakis,*

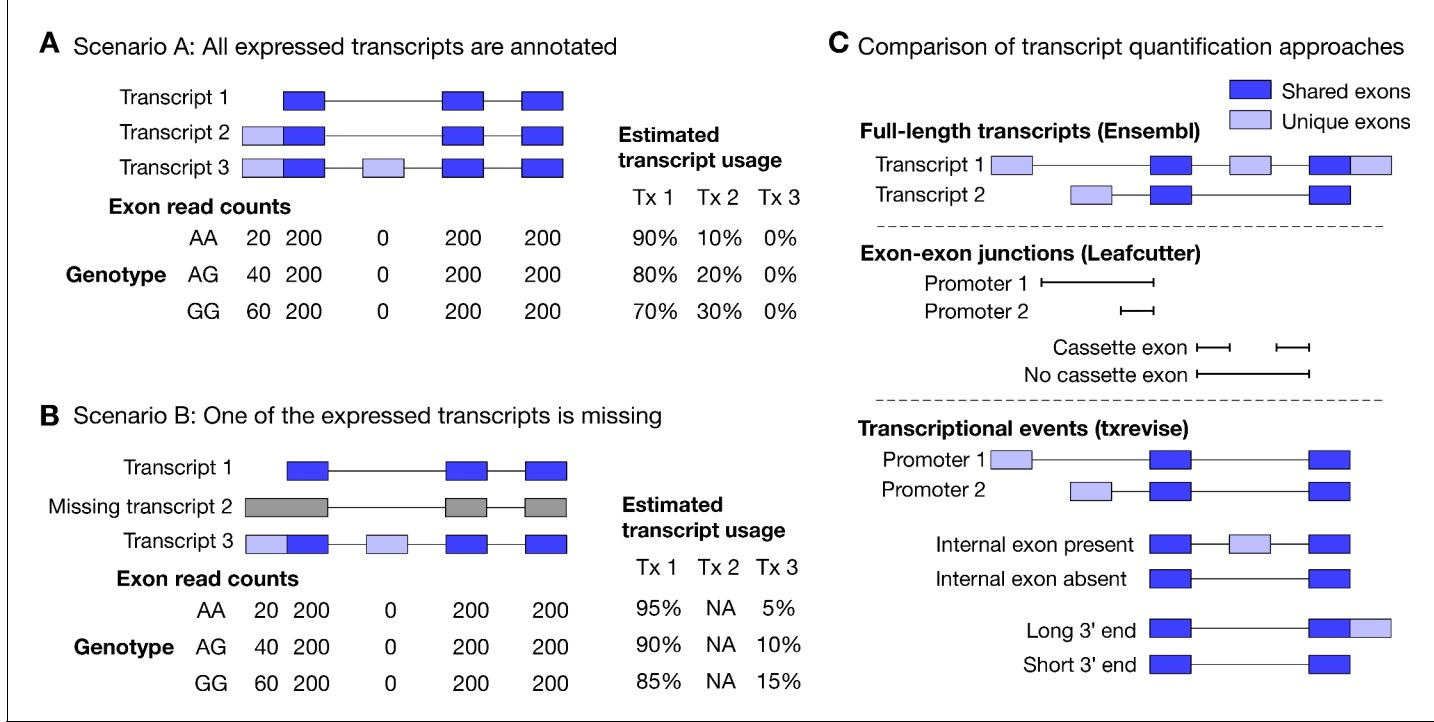

**Figure 1.** Challenges of quantifying transcript usage from RNA-seq data. Transcript quantification seeks to estimate the most likely configuration of *known* transcripts that best explains observed read counts supporting the inclusion of each exon. (**A**) In scenario A, each copy of the G allele increases the usage of transcript 2 by 10%. Since both expressed transcripts (transcript 1 and transcript 2) are annotated, we successfully detect the change and conclude that the G allele increases the expression of the proximal promoter of the gene. (**B**) In scenario B, each copy of the G allele still increases the usage of transcript 2 by 10%. However, since transcript 2 is missing from the annotations, reads originating from transcript 2 are now falsely assigned to transcript 3. Since transcript 3 also contains alternative second exon, we now falsely conclude that in addition to promoter usage, the G allele is also associated with increased inclusion of exon 2, even though there are no reads mapping to exon 2. Furthermore, the magnitude of the genetic effect is underestimated, because the reads assigned to transcript 3 are assumed to be evenly distributed across the promoter and the alternative exon. (**C**) Top panel: Two hypothetical transcripts that differ from each other at the promoter, at an internal exon and at the 3' end. Middle panel: Leafcutter uses reads mapping to exon-exon junctions to identify alternatively excised introns. Bottom panel: txrevise uses the exons shared between transcripts (dark blue) as a scaffold to construct three independent transcriptional events from the two original transcripts.

DOI: https://doi.org/10.7554/eLife.41673.003

The following figure supplements are available for figure 1:

**Figure supplement 1.** Prevalence of truncated transcripts in the Ensembl database.
DOI: https://doi.org/10.7554/eLife.41673.004

**Figure supplement 2.** Extending truncated transcript annotations with txrevise.
DOI: https://doi.org/10.7554/eLife.41673.005

**Figure supplement 3.** Identifying groups of transcripts that share the most exons.
DOI: https://doi.org/10.7554/eLife.41673.006

**Figure supplement 4.** Filling in alternative internal exons for promoter and 3' end events.
DOI: https://doi.org/10.7554/eLife.41673.007

*2015*), they do not explicitly reveal the transcriptional mechanism (promoter usage, alternative splicing or 3' end usage) underlying the genetic associations. The most successful approach to differentiate between distinct transcriptional mechanisms has been 'event-level' analysis where reference transcripts are split into independent events (e.g. promoters, splicing events and 3' ends) whose expressions is then quantified using standard transcript quantification methods (*Figure 1C*). This approach was pioneered by MISO (*Katz et al., 2010*) and was recently used to identify promoter usage QTLs in the GEUVADIS dataset (*Richards et al., 2017*). Despite its success, MISO covers only a subset of promoter events (alternative first exons) and its event annotations have not been updated since it was first published. Thus, there is a need for a method that is able to detect a comprehensive set of promoter, splicing and 3' end usage QTLs in an uniform manner.

In this study, we re-analysed RNA-seq data from human induced pluripotent stem cell-derived macrophages (IPSDMs) exposed to three inflammatory stimuli (18 hr IFNɣ stimulation, 5 hr *Salmonella* infection and IFNɣ stimulation followed by *Salmonella* infection) (*Alasoo et al., 2018*). We also collected a new dataset of IPSDMs from 70 individuals stimulated with acetylated LDL (acLDL) for 24 hr. We mapped genetic associations at the level of total gene expression, full-length transcript usage and exon-exon junction usage in each experimental condition. In addition to existing quantification methods, we also developed a complementary approach (txrevise) that stratifies reference transcript annotations into independent promoter, splicing and 3' end events. Using txrevise, we found that promoter and 3' end usage QTLs constituted 55% of detected tuQTLs, exhibited distinct genetic architectures from canonical expression or splicing QTLs, and often colocalised with complex trait associations. Promoter usage QTLs were also 50% more likely to be context-specific than canonical splicing QTLs. Thus, context-specific regulation of promoter usage might be a previously underappreciated molecular mechanism underlying complex trait associations.

## Results

### Quantifying transcript usage in stimulated macrophages

We analysed RNA-seq data from human induced pluripotent stem cell (iPSC)-derived macrophages exposed to three inflammatory stimuli (18 hr IFNɣ stimulation, 5 hr *Salmonella* infection, and IFNɣ stimulation followed by *Salmonella* infection) and one metabolic stimulus (24 hr acLDL stimulation). While the gene expression analysis of the IFNɣ+*Salmonella* dataset from 84 individuals has previously been described (*Alasoo et al., 2018*), the acLDL data from 70 individuals was newly generated for the current study. The acLDL dataset allowed us to assess how our results generalise to weaker, non-inflammatory stimuli. Both datasets included independent unstimulated control samples (denoted as 'naive' and 'Ctrl'). In each condition, we quantified gene expression and transcript usage using the following established quantification approaches: (i) gene-level read count quantified with featureCounts (*Liao et al., 2014*), (ii) full-length transcript usage quantified with Salmon (*Patro et al., 2017*) (*Figure 1C*), and (iii) exon-exon junction usage quantified with Leafcutter (*Li et al., 2018*) (*Figure 1C*).

Inspired by event level analysis proposed by MISO (*Katz et al., 2010*; *Richards et al., 2017*), we also developed a complementary approach (txrevise) to stratify reference transcript annotations into independent promoter, splicing and 3' end events. To achieve this, txrevise identifies constitutive exons shared between all transcripts of a gene and uses those to assign non-constitutive exons to promoter, internal exon or 3' end events (*Figure 1C*). Since up to 60% of the transcripts annotated by Ensembl (*Zerbino et al., 2018*) are truncated at the 5' or 3' end (*Figure 1—figure supplement 1*), txrevise extends truncated transcripts by copying over exons from the longest transcript of the gene (*Figure 1—figure supplement 2*). This step eliminates a large number of implausible alternative promoter and 3' end events that lack experimental evidence. To make the approach suitable for genes with non-overlapping transcripts, we also select a subset of transcripts that share the largest number of exons between them (*Figure 1—figure supplement 3*). Finally, to ensure that the new alternative promoter and 3' end events do not capture splicing changes, txrevise masks alternative exons in promoter and 3' end events that are not the first or last exons (*Figure 1—figure supplement 4*). Although this means that some splicing events near the promoters and 3' ends of the genes may remain undetected by txrevise, it is a trade-off that improves the overall interpretability of txrevise tuQTLs. The R package as well as custom transcriptional events constructed by txrevise are available from GitHub (https://github.com/kauralasoo/txrevise; *Alasoo, 2018a*).

### Genetic effects on transcript usage

Depending on the experimental condition and quantification method, we detected between 1500 and 3500 QTLs at a 10% false discovery rate (FDR) (*Figure 2A*). Leafcutter consistently detected the lowest number of QTLs per condition, while txrevise detected approximately 30% more associations than other methods (*Figure 2A*), 55% of which affected promoter or 3' end usage instead of internal exons (*Figure 2—figure supplement 1*). However, this increase in QTLs can be partially explained by the fact that txrevise detected multiple associations for ~24% of the genes while the full-length tuQTL analysis was limited to single lead association per gene (*Figure 2—figure supplement 2*,

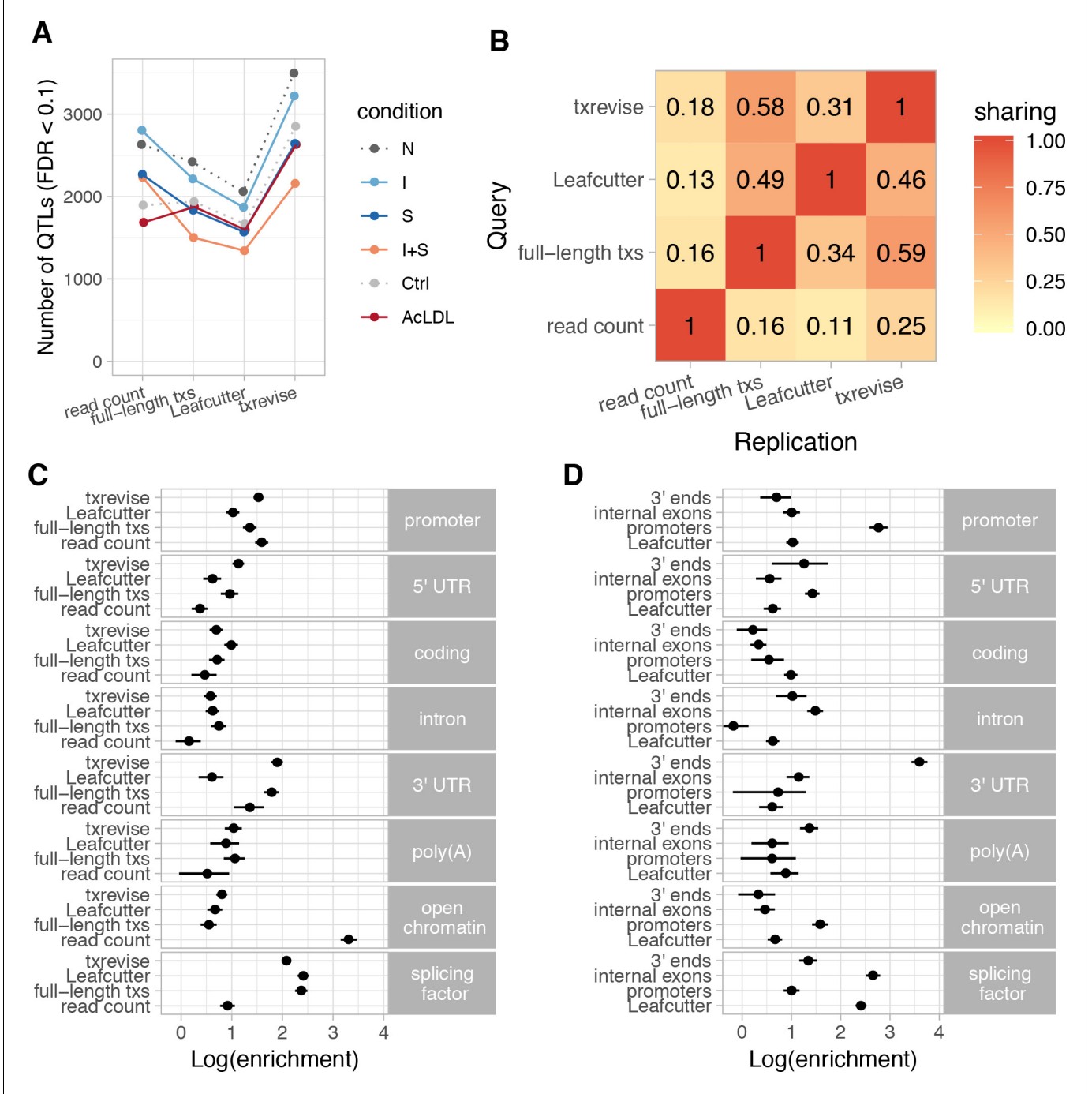

**Figure 2.** Diversity of QTLs detected by different quantification methods. In panels A-C, all txrevise QTLs from promoters, internal exons and 3' ends have been pooled to facilitate comparison with eQTLs as well as Leafcutter and full-length transcript usage QTLs. (**A**) Number of QTLs detected by read count, full-length transcript usage, Leafcutter and txrevise methods in each condition (N, naive; I, IFNɣ; S, *Salmonella*; I + S, IFNɣ+*Salmonella*; Ctrl; AcLDL) at 10% FDR. The number of QTLs detected by Leafcutter and txrevise are reported at the level of independent events (intron clusters or promoters/internal exons/3' ends) and can include multiple QTLs per gene (*Figure 2—figure supplement 2*). The quantile-quantile plots are presented in *Figure 2—figure supplement 3*. (**B**) Sharing of QTLs detected by four quantification methods. The numbers on the heatmap show the fraction of QTLs detect by one method that were replicated by each of the three other methods ($r^2$ >0.8 between lead variants). Only QTLs with FDR < 0.01 were included in the analysis. (**C**) Enrichment of genomic annotations at QTLs detected by the four quantification methods. (**D**) Comparison of Leafcutter tuQTLs to promoter, internal exon and 3' end usage QTLs detected by txevise. Genomic annotations used for enrichment analysis: *promoter* - promoter flanking regions (−2000 bp to +200 bp); *5' UTR, coding, intron, 3' UTR* - corresponding regions extracted from Ensembl transcripts; *poly(A)* - experimentally determined polyadenylation sites (±25 bp) (*Gruber et al., 2016*); *open chromatin* - open chromatin regions from macrophages

*Figure 2 continued on next page*

*Figure 2 continued*

(*Alasoo et al., 2018*); *splicing factor* - experimentally determined binding sites of splicing factors detected by eCLIP (*Van Nostrand et al., 2017*). The points on panels C and D show the natural logarithm of enrichment for each annotation and the lines represent the 95% confidence intervals from fgwas (*Pickrell, 2014*).

DOI: https://doi.org/10.7554/eLife.41673.008

The following figure supplements are available for figure 2:

**Figure supplement 1.** Diversity of transcript usage QTLs.
DOI: https://doi.org/10.7554/eLife.41673.009
**Figure supplement 2.** Fraction of genes with multiple independent tuQTLs detected by Leafcutter and txrevise.
DOI: https://doi.org/10.7554/eLife.41673.010
**Figure supplement 3.** Quantile-quantile plots of the QTLs detected by the four quantification methods.
DOI: https://doi.org/10.7554/eLife.41673.011
**Figure supplement 4.** Genetics of transcript usage of the *IRF5* gene.
DOI: https://doi.org/10.7554/eLife.41673.012
**Figure supplement 5.** Example of an apparent tuQTL caused by a strong eQTL.
DOI: https://doi.org/10.7554/eLife.41673.013
**Figure supplement 6.** Simulated promoter usage QTL for the *RNF220* gene leads to a false positive association at the 3' end.
DOI: https://doi.org/10.7554/eLife.41673.014
**Figure supplement 7.** Shared genetic effect on promoter usage and chromatin accessibility at the promoter of *HDLBP*.
DOI: https://doi.org/10.7554/eLife.41673.015

*Figure 2—figure supplement 3*). Some of these additional QTLs are likely to represent independent causal variants, such as the three independent tuQTLs detected for the *IRF5* gene (*Figure 2—figure supplement 4*) while others could be explained by technical biases such as large gene expression QTL (eQTL) effects (*Figure 2—figure supplement 5*) or positional biases in the RNA-seq data (*Figure 2—figure supplement 6*). Alternatively, additional associations could also be caused by transcriptional coupling where promoter or 3' end choice directly influences the splicing of an internal exon or *vice versa* (*Anvar et al., 2018*; *Bentley, 2014*).

Different quantification methods may be biased towards discovering events with specific genomic properties, which is not captured by the number of QTLs detected. To address this, we quantified how often the lead QTL variants (FDR < 0.01) from different methods were in high linkage disequilibrium (LD) ($r^2$ >0.8) with each other (see Materials and methods). Consistent with previous reports that tuQTLs are largely independent from eQTLs (*Li et al., 2016*), we found that only 11–25% of the lead variants detected at the read count level replicated at the transcript level ($r^2$ >0.8, irrespective of the replication p-value), independent of which quantification method was used (*Figure 2B*). In contrast, ~50% of the Leafcutter QTLs were also detected by txrevise or full-length transcript usage approaches. Similarly, the tuQTLs detected by txrevise and full-length transcript usage quantification were in high LD more than 60% of the time (*Figure 2B*). Finally, we found that while 44% of the txrevise internal exon QTLs were in high LD with Leafcutter QTLs, this decreased to ~20% for promoter and 3' end QTLs (*Figure 2—figure supplement 1*), suggesting that Leafcutter is less suited to capture those events. Thus, different quantification approaches appear to capture complementary sets of genetic associations.

## Genomic properties of transcript usage QTLs

To characterise the genetic associations detected by different quantification methods, we compared the relative enrichments of the identified QTLs across multiple genomic annotations. We constructed genomic tracks for eight annotations: open chromatin measured by ATAC-seq (*Alasoo et al., 2018*), promoter flanking regions (−2000 bp to +200 bp), 5' UTRs, coding sequence (CDS), introns, 3' UTRs, polyadenylation sites (*Gruber et al., 2016*), and eCLIP-binding sites for RNA-binding proteins involved in splicing regulation (splicing factors) (*Van Nostrand et al., 2017*). We then used the hierarchical model implemented in fgwas (*Pickrell, 2014*) to estimate the enrichment of each genomic annotation among the QTLs detected by each quantification method. Consistent with the limited overlap between eQTLs and tuQTLs (*Figure 2B*), we found that eQTLs were strongly enriched in sites of open chromatin (*Figure 2C*; log enrichment of 3.31, 95% CI [3.15, 3.47]), whereas all transcript-level QTLs were enriched at the binding sites of splicing factors detected by eCLIP

(*Figure 2C*, mean log enrichment of 2.29). Importantly, when all txrevise tuQTLs were pooled, the enrichment patterns were broadly similar to tuQTLs detected by full-length Ensembl transcripts (*Figure 2C*). This suggests that txrevise events and full-length transcripts capture similar genetic associations but txrevise facilitates more accurate identification of the underlying transcriptional event (i.e. promoter, internal exon or 3' end usage) (*Figure 2B*). Finally, compared to Leafcutter, full-length transcript usage and txrevise QTLs were both more strongly enriched at 3' UTRs (*Figure 2C*, mean log enrichment of 1.85), suggesting that they capture changes in 3' UTR length that do not manifest at the level of junction reads and are thus missed by Leafcutter.

To compare different types of transcriptional events, we repeated the fgwas analysis on the promoter, internal exon and 3' end QTLs detected by txrevise as well as Leafcutter splicing QTLs. We found that Leafcutter and internal exon QTLs showed broadly similar enrichment patterns, with a strong enrichment at the binding sites of splicing factors (*Figure 2D*, mean log enrichment of 2.53). In contrast, promoter and 3' end usage QTLs were specifically enriched at promoters (*Figure 2D*; log enrichment of 2.76, 95% CI [2.59, 2.95]) and 3' UTRs (*Figure 2D*; log enrichment of 3.60, 95% CI [3.43, 3.76]), respectively (*Figure 2D*), and showed only a modest enrichment at the binding sites of splicing factors (*Figure 2D*; mean log enrichment of 1.17). Compared to other events, promoter usage QTLs were relatively more enriched in open chromatin regions (log enrichment of 1.58, 95% CI [1.42, 1.74]). Thus, promoter usage, splicing and 3' end usage appear to be regulated by largely independent sets of genetic variants enriched in distinct genomic regions.

Motivated by the enrichment of promoter usage QTLs in open chromatin regions (*Figure 2D*), we analysed chromatin accessibility QTLs that we previously identified in a subset of 41 individuals of the same study (*Alasoo et al., 2018*). We wanted to determine how often changes in promoter usage also manifest at the level of promoter accessibility. We found that 124/786 (15.8%) of the promoter usage QTLs were in high LD with at least one chromatin accessibility QTL ($r^2$ >0.9) compared to 10.2% of the internal exon and 3' end usage QTLs (Fisher's exact test p-value=$3.87 \times 10^{-5}$). These overlaps could correspond to both distal regulatory elements affecting promoter usage or direct changes in local promoter accessibility. To focus on local promoter accessibility, we further required the center of the accessible region to be no farther than 1000 bp from the closest promoter of the gene, leaving 46/786 (5.8%) promoter usage QTLs with a putative coordinated effect on promoter accessibility. One such example affecting promoter usage and promoter accessibility of the *HDLBP* gene is highlighted in *Figure 2—figure supplement 7*. However, larger studies with increased statistical power are needed to characterise the true extent of coordination between promoter accessibility and promoter usage.

## Colocalisation with complex trait associations

To assess the relevance of different QTLs for interpreting complex trait associations, we performed statistical colocalisation analysis with GWAS summary statistics for 33 immune-mediated and metabolic traits and diseases (see Materials and methods). We found that 47 of 138 colocalised QTLs influenced total gene expression level (*Figure 3A*) (PP3+PP4 >0.8, PP4/PP3 >9; PP3, posterior probability of a model with two distinct causal variants; PP4, posterior probability of a model with one common causal variant). In contrast, the remaining 91 colocalised QTLs were associated with at least one of the transcript-level phenotypes (full-length transcript usage, txrevise or Leafcutter) but not with total gene expression (*Figure 3A*). Similarly, 44 of 91 transcript-level colocalisations were detected only by a single transcript quantification approach (*Figure 3A*). An important caveat of this analysis is that it does not directly test if the colocalisations are specific to one quantification method or simply missed by others because of limited power. Thus, our estimates of method-specificity are likely to be inflated.

Finally, to quantify the relative contribution of promoter usage, splicing and 3' end usage to complex traits, we stratified the txrevise colocalisations by transcriptional event type. We found that 44 of 77 colocalised QTLs influenced internal exons and the rest regulated promoters and 3' ends (*Figure 3B*). We were able to replicate known associations between splicing of exon two in *CD33* and Alzheimer's disease (*Figure 3—figure supplement 1*) (*Malik et al., 2013*) and splicing of exon 13 in *HMGCR* and LDL cholesterol (*Figure 3—figure supplement 2*) (*Burkhardt et al., 2008*). Importantly, while half of the promoter and internal exon colocalisations were also detected by Leafcutter, only 1/10 3' end events were captured by Leafcutter, probably because these are less likely to manifest at the level of junction reads (*Figure 3B*).

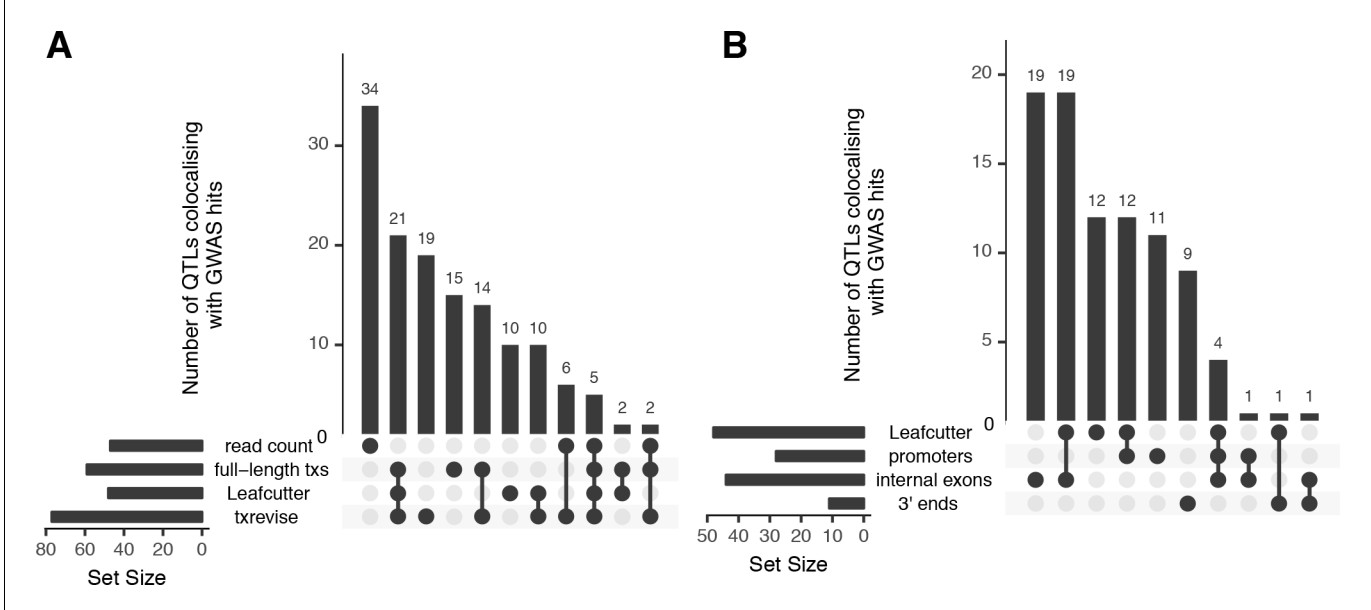

**Figure 3.** Overlap of colocalised gene-trait pairs detected by the four quantification methods across 33 complex traits. The UpSetR plot is an alternative to Venn diagrams for visualising intersection of multiple sets (*Conway et al., 2017*). The horizontal bars show the total number of colocalised trait-gene pairs detected by each quantification method. The dark circles indicate different patterns of sharing between the quantification methods and the vertical bars show how many gene-trait pairs followed a given sharing pattern. For example, in panel A, the first column shows that 34 colocalised gene-trait pairs were detected only at the total read count level but not at the transcript level. Similarly, the second column shows that 21 colocalised gene-trait pairs were detected by all transcript-level methods but not by total read count. (**A**) Sharing of colocalised gene-trait pairs between the four quantification methods. (**B**) Sharing of colocalised gene-trait pairs between Leafcutter and three independent txrevise event types (promoters, internal exons, 3' ends).

DOI: https://doi.org/10.7554/eLife.41673.016

The following figure supplements are available for figure 3:

**Figure supplement 1.** Colocalisation between *CD33* splicing QTL and GWAS hit for Alzheimer's disease.
DOI: https://doi.org/10.7554/eLife.41673.017
**Figure supplement 2.** Colocalisation between *HMGCR* splicing QTL and GWAS hit for LDL.
DOI: https://doi.org/10.7554/eLife.41673.018

## Condition-specificity of expression and transcript usage QTLs

Next, we explored how the genetic effects of eQTLs and tuQTLs varied in response to stimuli. To define response QTLs, we started with QTLs detected (FDR < 10%) in each of the four simulated conditions (I, S, I + S and acLDL) and used an interaction test to identify cases where the QTL effect size was significantly different between the simulated and corresponding naive condition (FDR < 10%). To exclude small but significant differences in effect size, we used a linear mixed model to identify QTLs where the interaction term explained more than 50% of the total genetic variance in the data (see Materials and methods). Although the fraction of QTLs that were response QTLs varied greatly between conditions (*Figure 4A*) and correlated with the number of differentially expressed genes (*Figure 4—figure supplement 1*) as previously reported (*Kim-Hellmuth et al., 2017*), we found that the fraction of response tuQTLs was relatively consistent between the four quantification methods (*Figure 4A*). While previous reports have highlighted that eQTLs are more condition-specific than tuQTLs (*Nédélec et al., 2016*), we found no clear pattern in our data with stronger stimuli (S and I + S) showing larger fraction of condition-specific eQTLs, and weaker stimuli (I, acLDL) showing smaller fraction of response eQTLs (*Figure 4A*) compared to tuQTLs. However, when we focussed on the transcriptional events detected by txrevise, we found that promoter usage QTLs were 50% more likely to be response QTLs than tuQTLs regulating either internal exons or 3' ends (*Figure 4B*) (Fisher's exact test combined p-value=$2.79 \times 10^{-6}$).

Finally, we assessed the condition-specificity of QTLs that colocalised with complex trait loci. We found that, on average, 12% of the GWAS colocalisations corresponded to response QTLs

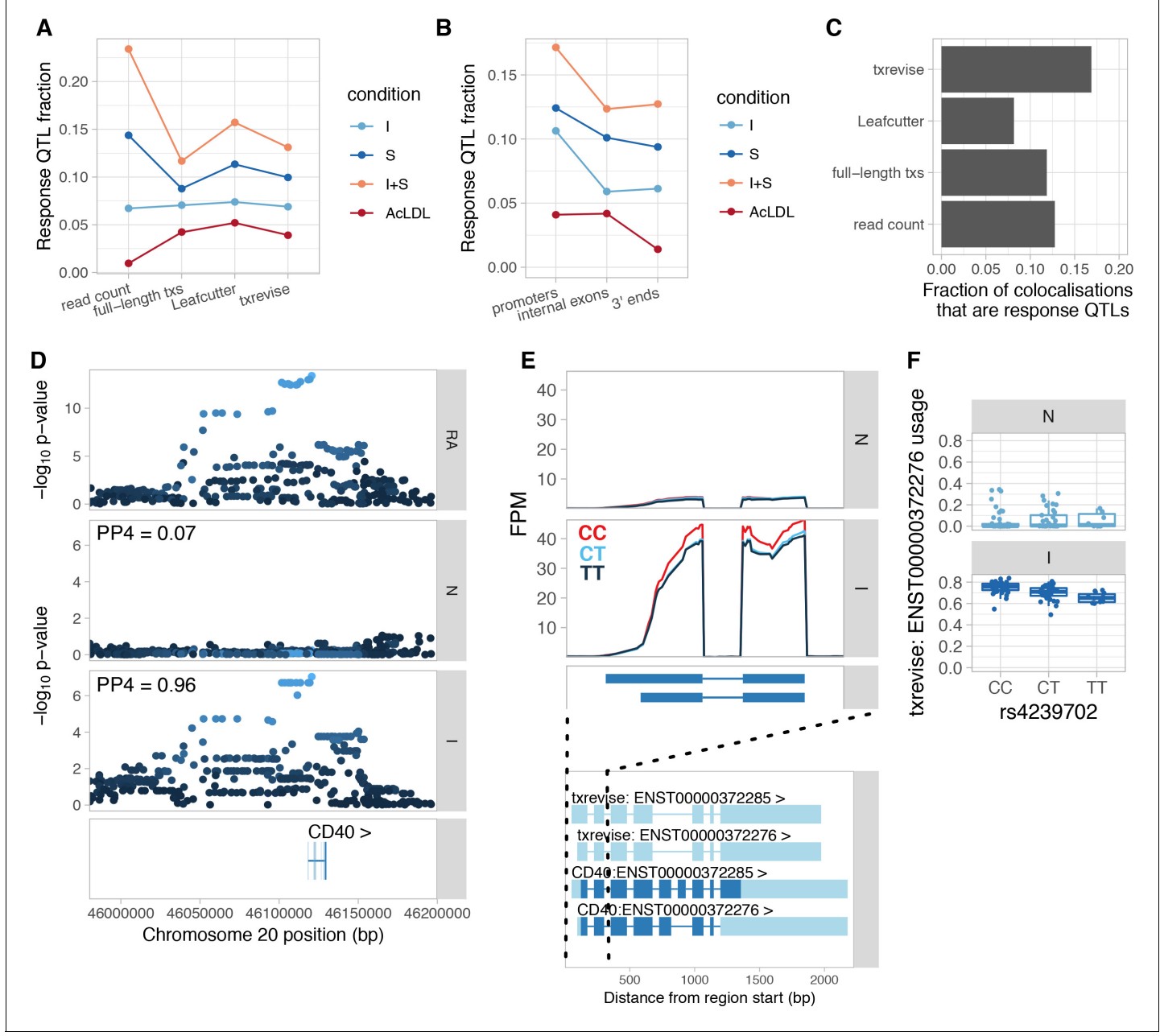

**Figure 4.** Condition-specificity of eQTLs and tuQTLs. (**A**) Fraction of all QTLs detected in each simulated condition that are response QTLs (FDR < 10% and more than 50% of the genetic variance explained by the interaction term). (**B**) Fraction of txrevise tuQTLs classified as response QTLs, stratified by the part of the gene that they influence (promoters, internal exons or 3' ends). (**C**) Fraction of GWAS colocalisations that are response QTLs. (**D**) Colocalisation between a GWAS hit for rheumatoid arthritis (RA) and IFNɣ-specific tuQTL at the *CD40* locus. PP4 represents the posterior probability from coloc (***Giambartolomei et al., 2014***) that the GWAS and QTL signals share a single causal variant. The dots are shaded according to $r^2$ from the lead RA GWAS variant. (**E**) Top panel: The lead GWAS variant (rs4239702) is associated with increased expression of the short 5' UTR of the *CD40* gene. Bottom panel: Ensembl annotations couple the short 5' UTR to skipped exon 6, but this is not supported by RNA-seq data (***Figure 4—figure supplement 2***). FPM, fragments per million. (**F**) Relative expression of the short 5' UTR stratified by the genotype of the lead GWAS variant. N, naive; I, IFNɣ; S, *Salmonella*; I + S, IFNɣ+*Salmonella*.

DOI: https://doi.org/10.7554/eLife.41673.019

The following figure supplements are available for figure 4:

**Figure supplement 1.** Relationship between differential expression and response eQTL count.

DOI: https://doi.org/10.7554/eLife.41673.020

**Figure supplement 2.** Genetics of *CD40* expression.

*Figure 4 continued on next page*

*Figure 4 continued*

DOI: https://doi.org/10.7554/eLife.41673.021

**Figure supplement 3.** Regulation of *CD40* promoter usage in response to 2 hr lipopolysaccharide (LPS) stimulation in primary macrophages.
DOI: https://doi.org/10.7554/eLife.41673.022

(*Figure 4C*). One example is an IFNγ-specific promoter usage QTL for the *CD40* gene that colocalises with a GWAS signal for rheumatoid arthritis (*Okada et al., 2014*). The alternative C allele of the rs4239702 variant is associated with increased usage of the transcript with the short 5' UTR (*Figure 4E,F*). This tuQTL was also visible at the absolute expression level of the two alternative promoters (*Figure 4—figure supplement 2*), but was missed by Leafcutter, because there is no change in junction reads. Although the variant was not significantly associated with total gene expression level (*Figure 4—figure supplement 2*), the two promoters contain the same start codon. As a result, the likely functional consequence of the *CD40* tuQTL is modulation of protein abundance. Although the same tuQTL was also detected at the full-length transcript usage level, the affected transcripts also differ from each other by alternatively spliced exon 6, making it challenging to interpret the result (*Figure 4E*). The preferential upregulation of the transcript with the short 5' UTR after exposure to an inflammatory stimulus is also supported by FANTOM5 capped analysis of gene expression (CAGE) data from primary macrophages (*Figure 4—figure supplement 3*) (*Baillie et al., 2017*).

## Discussion

We have performed a comprehensive analysis of the genetic determinants of transcript usage in human iPSC-derived macrophages exposed to four different stimuli. Our approach to stratify transcripts into individual events greatly improved the interpretability of molecular mechanisms underlying tuQTLs. Consequently, we were able to discover that 55% of the transcript-level associations affected promoter or 3' end usage and these variants were enriched in markedly different genomic features relative to canonical splicing QTLs. We also found that promoter usage QTLs were 50% more likely to be condition-specific than other transcriptional events and often colocalised with GWAS hits for complex traits. Thus, event-level analysis might be preferable over transcript-level analysis when the aim is to identify specific transcriptional changes underlying genetic associations.

We were able to link 6% of the promoter usage QTLs to coordinated changes in promoter accessibility. A likely reason for such a small overlap is limited statistical power in our chromatin accessibility dataset that contained only 41 individuals, leading us to miss many true effects on promoter accessibility. Alternatively, as other studies have suggested, promoter accessibility might not be an accurate proxy of activity and may merely be a prerequisite for transcription to take place (*Pliner et al., 2018*), but demonstrating this would require better powered datasets to confidently demonstrate lack of effect on promoter accessibility. There is a great potential to study this further in larger datasets that have profiled gene expression, chromatin accessibility or histone modifications in hundreds of individuals (*Chen et al., 2016*; *Kumasaka et al., 2019*).

Choosing the optimal quantification method for RNA-seq data is a challenging problem. The field of detecting and quantifying individual transcriptional changes from RNA-seq data has been developing rapidly. One of the most successful approaches has been the use of reads spanning exon-exon junctions to detect differential usage of individual exons within genes. In our study, we used Leafcutter to perform junction-level analysis, but other options are available such as JUM (*Wang and Rio, 2018*) or MAJIQ (*Vaquero-Garcia et al., 2016*). A key advantage of junction-level analysis is that it can discover novel exon-exon junctions and is thus well-suited for characterising rare or unannotated splicing events. On the other hand, changes in 5' and 3' UTR length are not captured by junction-level methods, because these events do not overlap exon-exon junctions. Changes in UTR length can only be detected by methods that consider all reads originating from alternative transcript ends such as MISO (*Katz et al., 2010*) or txrevise proposed here. MISO provides more fine-grained events that can differentiate between various types of splicing events. Txrevise, on the other hand, provides a more comprehensive catalog of promoter and 3' end events that can be continuously updated as reference annotations improve. A promising alternative to both of these methods is Whippet, which quantifies transcriptional events by aligning reads directly to the splice graph of the gene (*Sterne-Weiler et al., 2017*). Thus, no single approach is consistently superior to others

and characterizing the full spectrum of transcriptional consequences of genetic variation requires a combination of analytical strategies (*Odhams et al., 2017*; *Ongen and Dermitzakis, 2015*).

An important limitation of txrevise is that it is only able to quantify splicing events present in reference transcript databases. However, our approach can easily be extended by incorporating additional annotations such experimentally determined promoters from the FANTOM5 (*Forrest et al., 2014*) projects or alternative polyadenylation sites from the PolyAsite database (*Gruber et al., 2016*), as is done by QAPA (*Ha et al., 2018*). Another option might be to incorporate novel transcripts identified by transcript assembly methods such as StringTie (*Pertea et al., 2015*) into existing annotation databases. Nevertheless, since txrevise relies on Salmon for event-level quantification, it is still susceptible to some of the same limitations as full-length transcript quantification. Even though event-level analysis reduces the problem slightly, a positive transcript expression estimate does not guarantee that any specific exon is actually present in the transcript, especially if the transcript annotations are incomplete (*Figure 1B*) (*Soneson et al., 2018*). Secondly, large eQTL effects and positional biases in the RNA-seq data can occasionally lead to spurious changes in transcript usage (*Figure 2—figure supplements 5* and *6*). Therefore, it is important to visually confirm candidate transcriptional events using either base-level read coverage plots (*Alasoo, 2017*) or Sashimi plots (*Katz et al., 2015*) before embarking on follow-up experiments.

A key aim of QTL mapping studies is to elucidate the molecular mechanisms underlying complex trait associations. In our analysis, we found that over 50% of the genetic effects that colocalise with complex traits regulated transcript usage and did not manifest at the total gene expression level. Moreover, 42% of the transcript-level colocalisations affected promoter or 3' end usage instead of splicing of internal exons. Importantly, no single quantification method was able to capture the full range of genetic effects, confirming that different quantification approaches often identify complementary sets of QTLs (*Odhams et al., 2017*; *Ongen and Dermitzakis, 2015*). Thus, there is great potential to discover additional disease associations by re-analysing large published RNA-seq datasets such as GTEx (*Battle et al., 2017*) with state-of-the-art quantification methods.

## Materials and methods

### Cell culture and reagents

#### Donors and cell lines

Human induced pluripotent stem cells (iPSCs) lines from 123 healthy donors (72 female and 51 male) (*Supplementary file 1*) were obtained from the HipSci project (*Kilpinen et al., 2017*). Of these lines, 57 were initially grown in feeder-dependent medium and 66 were grown in feeder-free E8 medium. The cell lines were screened for mycoplasma by the HipSci project (*Kilpinen et al., 2017*). All samples for the HipSci project (*Kilpinen et al., 2017*) were collected from consented research volunteers recruited from the NIHR Cambridge BioResource (http://www.cambridgebioresource.org.uk). Samples were initially collected under ethics for iPSC derivation (REC Ref: 09/H0304/77, V2 04/01/2013), which require managed data access for all genetically identifying data. Later samples were collected under a revised consent (REC Ref: 09/H0304/77, V3 15/03/2013) under which all data, except from the Y chromosome from males, can be made openly available. The ethics approval was obtained from East of England - Cambridge East Research Ethics Committee. The iPSC lines used in this study are commercially available via the European Collection of Authenticated Cell Cultures. No new primary human samples were collected for this study.

The details of the iPSC culture, macrophage differentiation and stimulation for the IFNγ+*Salmonella* study have been described previously (*Alasoo et al., 2018*) (*Supplementary file 2*). Macrophages for the acLDL study were obtained from the same differentiation experiments.

#### AcLDL stimulation

Macrophages differentiated from a total of 71 iPSC lines were used for the acLDL stimulation. The final sample size was decided on the basis of similar gene expression and splicing QTL mapping studies performed previously (*Alasoo et al., 2018*; *Li et al., 2016*; *Nédélec et al., 2016*). Macrophages were grown in RPMI 1640 (Gibco) supplemented with 10% FBS (labtech), 2 mM L-glutamine (Sigma) and 100 ng/ml hM-CSF (R and D) at a cell density of 150,000 cells per well on a six-well plate. On day 6 of the macrophage differentiation, two wells of the six-well plate were exposed to

100 µg/ml human acLDL (Life Technologies) for 24 hr, whereas the other two wells were incubated in fresh RPMI 1640 medium without stimulation throughout this period.

For RNA extraction, cells were washed once with PBS and lysed in 300 µl of RLT buffer (Qiagen) per well of a six-well plate. Lysates from two wells were immediately pooled and stored at −80°C. RNA was extracted using a RNA Mini Kit (Qiagen) following the manufacturer's instructions and eluted in 35 µl nuclease-free water. RNA concentration was measured using NanoDrop, and RNA integrity was measured on Agilent 2100 Bioanalyzer using a RNA 6000 Nano Total RNA Kit.

## RNA sequencing and quality control

All RNA-seq libraries from the acLDL study were constructed manually using poly-A selection and the Illumina TruSeq stranded library preparation kit. The TruSeq libraries were quantified using Bioanalyzer and manually pooled for sequencing. The samples were sequenced on Illumina HiSeq 2000 using V4 chemistry and multiplexed at six samples/lane. The control and acLDL stimulated RNA samples from a single donor were always sequenced in the same experimental batch. Sample metadata is presented in *Supplementary file 2*. RNA-seq reads from both studies were aligned to the GRCh38 reference genome and Ensembl 87 transcript annotations using STAR v2.4.0j (*Dobin et al., 2013*). Subsequently, VerifyBamID v1.1.2 (*Jun et al., 2012*) was used to detect and correct any sample swaps between donors. Two samples from one donor (HPSI0513i-xegx_2) were excluded from downstream analysis, because they appeared to be outliers on the principal component analysis (PCA) plot of the samples.

## Quantifying gene and transcript expression

We used four alternative strategies to quantify transcription from RNA-seq data: (i) gene-level read count quantified with featureCounts (*Liao et al., 2014*), (ii) full-length transcript usage quantified with Salmon (*Patro et al., 2017*) (*Figure 1C*), (iii) promoter, internal exon and 3' end usage quantified with txrevise, and (iv) exon-exon junction usage quantified with Leafcutter (*Li et al., 2018*).

### Gene-level read counts

We used featureCounts v1.5.0 (*Liao et al., 2014*) to count the number of uniquely mapping fragments overlapping transcript annotations from Ensembl 87. We excluded short RNAs and pseudogenes from the analysis leaving 35,033 unique genes of which 19,796 were protein coding. Furthermore, in both IFNɣ+*Salmonella* and acLDL dataset, we used only genes with mean expression in at least one of the conditions greater than one transcripts per million (TPM) (*Wagner et al., 2012*) in all downstream analyses. This resulted in 12,660 and 12,103 genes included for analysis in the IFNɣ+*Salmonella* and acLDL datasets, respectively. We quantile-normalised the data and corrected for sample-specific GC content bias using the conditional quantile normalisation (cqn) (*Hansen et al., 2012*) R package as recommended previously (*Ellis et al., 2013*).

### Full-length transcript usage

We downloaded the FASTA files with messenger RNA (mRNA) and non-coding RNA sequences from the Ensembl website (version 87). We concatenated the two files and used salmon v0.8.2 (*Patro et al., 2017*) with '–seqBias –gcBias –libType ISR' options to quantify the expression level of each transcript. We used tximport (*Soneson et al., 2015*) package to import the expression estimates into R and calculated the relative expression of each transcript by dividing the TPM expression estimate of each transcript with the sum of the expression estimates of all transcripts of the gene.

### Quantifying transcriptional events with txrevise

We downloaded exon coordinates for all Ensembl 87 transcripts using the makeTxDbFromBiomart function from the GenomicFeatures (*Lawrence et al., 2013*) R package. We also downloaded metadata for these transcripts using the biomart (*Durinck et al., 2005*) R package. Finally, we extracted transcript tags from the GTF file downloaded from the Ensembl website using the extractTranscript-Tags.py script available from the txrevise repository (https://github.com/kauralasoo/txrevise). This step was necessary, because Ensembl contains a large number of truncated transcripts (marked with

cds_start_NF or cds_end_NF tags) (*Figure 1—figure supplement 1*), but this information is not present in biomart.

We developed the txrevise R package to pre-process transcript annotations prior to quantification. First, we extended all truncated protein coding transcripts using exons from the longest annotated transcript of the gene that was part of the GENCODE Basic gene set (*Figure 1—figure supplement 2*). We also performed the same step on transcripts annotated in Ensembl as retained_intron, processed_transcript or nonsense_mediated_decay, because they often ended abruptly in the middle of the exons and were unlikely to correspond to true transcription start and end sites.

Next, we focused on splitting full-length transcripts into alternative promoters, internal exons and 3' ends. However, some genes contained either non-overlapping transcripts or very short transcripts that complicated this process. Thus, for each gene we first identified a group of transcripts that shared the largest number of exons with each other. We then used the shared exons as a scaffold to construct three types of independent transcriptional events: alternative promoters, internal exons and 3' ends (group 1) (*Figure 1—figure supplement 3*). We also repeated this process on a second group of transcripts that shared the second-most exons with each other (group 2) (see *Figure 1—figure supplement 3* for illustration). Thus, the original transcripts from each gene were split into up to six sets of transcriptional events (two sets of alternative promoters, internal exons and 3' ends). Next, to ensure that the new alternative promoter and 3' end events did not capture splicing changes, we masked all alternative exons that were not the first or last exons (*Figure 1—figure supplement 4*). We applied this step only to alternative promoter and 3' end events and not to internal exon events. This final step can optionally be skipped to discover more association at the expense of losing some interpretability, because a subset of the promoter and 3' end events might be tagging splicing changes. We used Salmon (*Patro et al., 2017*) with '–seqBias –gcBias –libType ISR' options to independently quantify the expression of each set of transcriptional events. Finally, we used tximport (*Soneson et al., 2015*) to import the event expression estimates into R and calculated the relative expression of each transcriptional event by dividing the TPM expression estimate of each event with the sum of the expression estimates of all events within the same group of transcripts. This normalisation was performed separately for each type of transcriptional event (promoters, internal exons and 3' ends) and also separately for the two groups of transcripts used for constructing the alternative events, ensuring that the normalized value always represented the relative usage of one transcriptional event compared to other events of the same type that shared the same scaffold.

### Quantifying intron excision ratios with Leafcutter

Finally, we used Leafcutter (*Li et al., 2018*) to quantify the relative excision frequencies of alternative introns. We used the spliced alignments from STAR as input to Leafcutter. We did not correct for reference mapping bias, because we wanted to be able to directly compare Leafcutter results with those from Salmon and there is no obvious way to correct for reference mapping bias in Salmon quantification. We used the default parameters of requiring at least 50 reads supporting each intron cluster and allowing introns of up to 500 kb in length.

## Mapping expression and transcript usage QTLs

### Preparing genotype data

We obtained imputed genotypes for all of the samples from the HipSci (*Kilpinen et al., 2017*) project. We used CrossMap v0.1.8 (*Zhao et al., 2014*) to convert variant coordinates from GRCh37 reference genome to GRCh38. Subsequently, we filtered the VCF file with bcftools v.1.2 to retain only bi-allelic variants (both SNPs and indels) with IMP2 score >0.4 and minor allele frequency (MAF) >0.05. We created a separate VCF files for the IFNɣ+*Salmonella* study (84 individuals) and the acLDL study (70 individuals). The same VCF files were used for all downstream analyses and were imported into R using the SNPRelate R package (*Zheng et al., 2012*).

### Association testing

We used QTLTools (*Delaneau et al., 2017*) to map QTLs in two stages. First, we used the permutation pass with '–permute 10000 –grp-best' options to calculate the minimal lead variant p-value for each feature (gene, transcript or splicing event) in a ± 100 kb window around each feature. We included the first six principal components of the phenotype matrix as covariates in the QTL analysis.

The '–grp-best' option ensured that in case of transcript usage QTLs, the permutation p-values were corrected for the number of alternative transcripts, exon-exon junction or transcriptional events tested. For txrevise, we performed the permutations across the two groups of transcripts what were used for event construction. Quantile-quantile plots confirmed that the permutation p-values were well calibrated (*Figure 2—figure supplement 3*). Finally, we performed Benjamini-Hochberg FDR correction on the permutation p-values to identify QTLs that were significant at the 10% FDR level. For gene expression and full-length transcript usage QTLs this approach identified at most one lead variant per gene. For txrevise analysis we report up to three lead variants per gene: one for each independent transcriptional event type (promoters, internal exons, 3' ends) (*Figure 2—figure supplement 3*). If the two groups of events identified different genetic associations, we report only the one lead variant that had the smallest permutation p-value across groups. Leafcutter analysis sometimes also identified multiple associations per gene if there were multiple independent intron clusters within those genes (*Figure 2—figure supplements 2* and *3*). Finally, we note that additional secondary associations could be discovered by performing conditional analysis on the QTLs that have already been detected, but we did not do that analysis.

Secondly, we used the nominal pass to calculate nominal association p-values in a ± 500 kb *cis* window around each feature. We used a larger *cis* window for the nominal pass to ensure that we did not have missing data in the colocalisation analysis (see below), where we used the ±200 kb *cis* window around each lead QTL variant. However, the colocalisation analysis was still based on the lead QTL variants identified in the ±100 kb window. Thus, even if a stronger QTL lead variant was detected in the ±500 kb window, this was not used for any downstream analysis.

## QTL replication between quantification methods

To compare the QTLs detected by different quantification methods, we estimated the fraction of QTL lead variants detected by each method that were replicated by the other methods. Since read count and full-length transcript usage analysis were performed at the gene level, we decided to perform the replication analysis at the gene level as well. Because txrevise and Leafcutter quantified multiple events per gene and sometimes detected multiple independent QTLs (*Figure 2—figure supplement 2*), we picked the lead variant with the smallest p-value across all of the events quantified for a given gene as the gene-level lead variant. For each pairwise comparison of quantification methods, we first identified all lead variant-gene pairs with FDR < 0.01 detected by the query method. Subsequently, we extracted the lead variants for the same genes detected by the replication method and estimated the fraction of those that were in high LD ($r^2$ >0.8) with each other. We then repeated this analysis for all pairs of quantification methods. Note that this measure is not necessarily symmetric between the quantification methods and also depends on the statistical power of each method. Since Leafcutter had lower statistical power than other methods on our dataset, it also replicated smaller fraction of QTLs detected by the other methods. In contrast, ~50% of the Leafcutter QTLs were replicated by txrevise and full-length transcript usage (*Figure 2B*).

We acknowledge that our definition of replication ignores the direction of the effect of the genetic variant on gene expression or transcript usage. For example, if Leafcutter detects a genetic variant that is associated with increased inclusion of an exon in a gene and txrevise detects that the same variant is associated with decreased inclusion of the same exon in the same gene, we would still consider it to be a 'successful' replication. However, in practice it is difficult to map Leafcutter events to specific Ensembl transcripts or txrevise events, especially if Leafcutter includes novel exon-exon junctions not present in the Ensembl database. Furthermore, comparing the effect size direction between eQTLs and tuQTLs is not possible, because any variant that is associated with increased usage of one transcript is by definition also associated with decreased usage of some other transcripts of the same gene.

## QTL enrichment in genomics annotations

### Constructing genomic annotations

#### Gene features

We downloaded transcript annotations from Ensembl version 87 (*Zerbino et al., 2018*) using the GenomicFeatures (*Lawrence et al., 2013*) R package. We retained only protein coding transcripts and used fiveUTRsByTranscript, threeUTRsByTranscript, cdsBy, intronsByTranscript and promoters

functions to extract 5' UTRs, 3' UTRs, coding sequences, introns and promoters, respectively. We defined promoters as sequences 2000 bp upstream and 200 bp downstream of the annotated transcription start sites.

### Polyadenylation sites

We downloaded the coordinates of experimentally determined human polyadenylation sites from the PolyASite database (version r1.0) (*Gruber et al., 2016*). After converting the coordinates to the GRCh38 reference genome with CrossMap (*Zhao et al., 2014*), we extended each polyadenylation site to ±25 bp from the center of the site.

### Chromatin accessibility

We downloaded the coordinates of accessible chromatin regions in macrophages across four conditions (N, I, S, I + S) from our previous study (*Alasoo et al., 2018*). Specifically, we downloaded the ATAC_peak_metadata.txt.gz file from Zenodo (https://doi.org/10.5281/zenodo.1170560).

### RNA-binding proteins

We downloaded processed eCLIP (*Van Nostrand et al., 2016*) peak calls for 93 RNA binding proteins (RBPs) (*Van Nostrand et al., 2017*) from the ENCODE web site (https://www.encodeproject.org). Each protein was measured in two biological replicates, resulting in 186 sets of peaks. We only used data from the K562 myelogenous leukemia cell line. We further used Supplementary Table 1 from (*Van Nostrand et al., 2017*) to identify a subset of 29 RBPs that have previously been implicated in splicing regulation, five factors that have been implicated in 3' end processing and two factors (SRSF7 and HNRNPK) that have been implicated in both. Within each group (splicing, 3' end processing and both), we first removed all peaks that were detected only once and then merged all peaks into a single genomic annotation.

### Enrichment analysis

We used fgwas v0.3.6 (*Pickrell, 2014*) with the '-fine' option to identify the genomic annotations in which different types of QTLs were enriched. We converted QTLtools p-values to z-scores using the stats.norm.ppf(p/2, loc = 0, scale = 1) function from SciPy (*Jones et al., 2001*), where p is the nominal p-value from QTLtools. The sign of the z-score was determined based on the sign of the QTL effect size. We included all genomic annotations into a joint fgwas model using the '-w' option. For the enrichment analysis, we used QTLs from the naive condition only, but we found that the enrichments patterns were very similar in all stimulated conditions.

## Context-specificity of expression and transcript usage QTLs

To identify response QTLs, we started with QTLs detected (FDR < 10%) in each of the four stimulated conditions (I, S, I + S and acLDL) and used an interaction test to identify cases where the QTL effect size was significantly different between one of the stimulated and corresponding naive condition (FDR < 10%). We performed this test separately for each of the four stimulated condition (I, S, I + S and AcLDL). Furthermore, to take advantage of our profiling of gene expression in overlapping set of donors in the stimulated and naive conditions, we also included the cell line as a random effect and fitted a linear mixed model using the lme4 (*Bates et al., 2015*) package. Specifically, for each phenotype and lead variant pair, we used the anova function to compare the following two models:

H0: phenotype ~genotype + condition + (1|donor)

H1: phenotype ~genotype + condition+condition:genotype + (1|donor)

where (1|donor) denotes the donor-specific random effect. We obtained the p-value of rejecting the null hypothesis and used the p.adjust function to identify phenotype and lead variant pairs that were significant at 10% Benjamini-Hochberg FDR.

For some QTLs, we noticed that although the interaction test p-value was significant, the difference in the effect size between the two conditions was very small. To identify response QTLs with large effect size differences between naive and stimulated conditions, we turned to variance component analysis. Specifically, for the same phenotype and lead variant pairs tested above, we also fitted the following linear mixed model:

phenotype ~ (1|genotype) + (1|condition) + (1|condition:genotype)

where genotype, condition and the interaction between the two were all fitted as random effects. We then quantified the variance explained by each of the three components using the VarCorr function form the lme4 package. Finally, we calculated the variance explained by the interactions term relative to the total genetic variance:

$$\sigma^2_{relative} = \sigma^2_{interaction} / (\sigma^2_{interaction} + \sigma^2_{genotype})$$

We defined response QTLs as those with FDR < 10% from the interaction test and $\sigma^2_{relative} > 0.5$ from the variance component analysis. Although fitting genotype as a random effect in this way is suboptimal because it ignores the expected linear relationship between the alternative allele dosage and phenotype, we empirically found that filtering both on the p-value of the interaction test as well as $\sigma^2_{relative}$ was effective at identifying QTLs with large effect size differences between conditions.

## Overlap with genome-wide association studies

### Summary statistics

We obtained full summary statistics for ten immune-mediated disorders: inflammatory bowel disease (IBD) including ulcerative colitis (UC) and Crohn's disease (CD) (*Liu et al., 2015*), Alzheimer's disease (AD) (*Lambert et al., 2013*), rheumatoid arthritis (RA) (*Okada et al., 2014*), systemic lupus erythematosus (SLE) (*Bentham et al., 2015*), type one diabetes (T1D) (*Onengut-Gumuscu et al., 2015*), schizophrenia (SCZ) (Schizophrenia Working Group of the *Schizophrenia Working Group of the Psychiatric Genomics Consortium, 2014*), multiple sclerosis (MS) (*Beecham et al., 2013*), celiac disease (CEL) (*Trynka et al., 2011*) and narcolepsy (NAR) (*Faraco et al., 2013*). We also obtained summary statistics for type two diabetes (T2D) (*Morris et al., 2012*), cardiovascular disease (CAD) (*Nelson et al., 2017*; *Nikpay et al., 2015*) and myocardial infarction (MI) (*Nikpay et al., 2015*). Finally, we obtained summary statistics for 20 cardiometabolic traits from a recent meta-analysis (*Iotchkova et al., 2016*). Summary statistics for T1D, CEL, IBD, RA, AD, MS and SLE were downloaded in 2015. SCZ, T2D and NAR were downloaded in 2016. T2D summary statistics were converted from GRCh36 to GRCh37 coordinates using the LiftOver tool, all the other summary statistics already used GRCh37 coordinates.

### Colocalisation analysis

We used coloc v2.3–1 (*Giambartolomei et al., 2014*) to test for colocalisation between gene expression and transcript usage QTLs and GWAS hits. We ran coloc on a 400 kb region centered on each lead eQTL and tuQTL variant that was less than 100 kb away from at least one GWAS variant with a nominal p-value$<10^{-5}$. We used the following prior probabilities: p1 = $10^{-4}$, p2 = $10^{-4}$ and p12 = $10^{-5}$. We then applied a set of filtering steps to identify a stringent set of eQTLs and tuQTLs that colocalised with GWAS hits. Similarly to a previous study (*Guo et al., 2015*), we first removed all cases where PP3+PP4 <0.8, to exclude loci where we were underpowered to detect colocalisation. We then required PP4/(PP3+PP4) >0.9 to only keep loci where coloc strongly preferred the model of a single shared causal variant driving both association signals over a model of two distinct causal variants. We excluded all colocalisation results from the MHC region (GRCh38: 6:28,510,120–33,480,577) because they were likely to be false positives due to complicated LD patterns in this region. We only kept results where the minimal GWAS p-value was $<10^{-6}$. Plots illustrating the sharing of colocalised GWAS signals by different quantification methods were made using UpSetR (*Conway et al., 2017*).

## Code availability

The Snakemake (*Köster and Rahmann, 2012*) files used for gene and transcript expression quantification, QTL mapping and colocalisaton are available from the project's GitHub repository (https://github.com/kauralasoo/macrophage-tuQTLs; copy archived at https://github.com/elifesciences-publications/macrophage-tuQTLs; *Alasoo, 2018b*). The same repository also contains R scripts that were used for all data analysis and figures. The txrevise R package is available from GitHub (https://github.com/kauralasoo/txrevise; copy archived at https://github.com/elifesciences-publications/txrevise; *Alasoo, 2018a*) and wiggleplotr R package that was used to make transcript read coverage plots is available from Bioconductor (http://bioconductor.org/packages/wiggleplotr/).

## Data availability

RNA-seq data from the acLDL stimulation study is available from ENA (PRJEB20734) and EGA (EGA S00001000876). RNA-seq data from the IFNɣ+*Salmonella* study is available from ENA (PRJEB18997) and EGA (EGAS00001002236). The imputed genotype data for HipSci cell lines is available from ENA (PRJEB11749) and EGA (EGAD00010000773). Processed data and QTL summary statistics are available from Zenodo: https://zenodo.org/communities/macrophage-tuqtls/.

## Acknowledgements

We thank Jeremy Schwartzentruber and Leopold Parts for their helpful comments on the manuscript. We thank WTSI DNA Pipelines and Cytometry Core Facility for their sequencing and flow cytometry services. We thank Elena Vigorito and Joanna MM Howson (Cardiovascular Epidemiology Unit) for their help with statistical analyses during the early phases of this project. This work was supported by the Wellcome Trust (WT098051) and the British Heart Foundation Cambridge Centre of Excellence (RE/13/6/30180). KA was supported by a PhD fellowship from the Wellcome Trust (WT099754/Z/12/Z), a postdoctoral fellowship from the Estonian Research Council (MOBJD67) and a grant from the Estonian Research Council (IUT34-4). The Cardiovascular Epidemiology Unit is supported by the UK Medical Research Council (MR/L003120/1), British Heart Foundation (RG/13/13/30194) and National Institute for Health Research [Cambridge Biomedical Research Centre at the Cambridge University Hospitals NHS Foundation Trust]. The views expressed are those of the authors and not necessarily those of the NHS, the NIHR or the Department of Health and Social Care. The iPSC lines were generated at the Wellcome Sanger Institute, under the Human Induced Pluripotent Stem Cell Initiative funded by a strategic award (WT098503) from the Wellcome Trust and Medical Research Council. We also acknowledge Life Science Technologies Corporation as the provider of cytotune. This work was carried out in part at the High-Performance Computing Center of University of Tartu.

## Additional information

### Competing interests

Daniel F Freitag: Since October 2015, Daniel F Freitag has been a full-time employee of Bayer AG, Germany. The other authors declare that no competing interests exist.

### Funding

| Funder | Grant reference number | Author |
|---|---|---|
| Wellcome | WT09805 | Kaur Alasoo<br>Julia Rodrigues<br>Daniel J Gaffney |
| British Heart Foundation | RG/13/13/30194 | John Danesh<br>Daniel F Freitag<br>Dirk S Paul |
| Estonian Research Council | MOBJD67 | Kaur Alasoo |
| Wellcome | WT099754/Z/12/Z | Kaur Alasoo |
| Estonian Research Council | IUT34-4 | Kaur Alasoo |
| Wellcome | WT098503 | Daniel J Gaffney |
| British Heart Foundation Cambridge Centre of Excellence | RE/13/6/30180 | John Danesh<br>Daniel F Freitag<br>Dirk S Paul |
| Medical Research Council | MR/L003120/1 | John Danesh<br>Daniel F Freitag<br>Dirk S Paul |

The funders had no role in study design, data collection and interpretation, or the decision to submit the work for publication.

## Author contributions
Kaur Alasoo, Conceptualization, Data curation, Software, Formal analysis, Funding acquisition, Investigation, Methodology, Writing—original draft, Writing—review and editing, Performed the macrophage differentiation experiments; Julia Rodrigues, Data curation, Investigation, Methodology, Performed the macrophage differentiation experiments, Optimised and performed the acLDL stimulation experiments; John Danesh, Supervision, Funding acquisition, Project administration; Daniel F Freitag, Conceptualization, Funding acquisition, Methodology, Project administration; Dirk S Paul, Resources, Supervision, Writing—original draft, Project administration, Writing—review and editing; Daniel J Gaffney, Conceptualization, Resources, Supervision, Funding acquisition, Methodology, Writing—original draft, Project administration, Writing—review and editing

## Author ORCIDs
Kaur Alasoo (iD) http://orcid.org/0000-0002-1761-8881
Dirk S Paul (iD) http://orcid.org/0000-0002-8230-0116

## Ethics
Human subjects: Human induced pluripotent stem cells (iPSCs) lines from 123 healthy donors (72 female and 51 male) (Supplementary file 1) were obtained from the HipSci project (Kilpinen et al., 2017). All samples for the HipSci project (Kilpinen et al., 2017) were collected from consented research volunteers recruited from the NIHR Cambridge BioResource (http://www.cambridgebioresource.org.uk). Samples were initially collected under ethics for iPSC derivation (REC Ref: 09/H0304/77, V2 04/01/2013), which require managed data access for all genetically identifying data. Later samples were collected under a revised consent (REC Ref: 09/H0304/77, V3 15/03/2013) under which all data, except from the Y chromosome from males, can be made openly available. The ethics approval was obtained from East of England - Cambridge East Research Ethics Committee. The iPSC lines used in this study are commercially available via the European Collection of Authenticated Cell Cultures. No new primary human samples were collected for this study.

## Decision letter and Author response
Decision letter https://doi.org/10.7554/eLife.41673.039
Author response https://doi.org/10.7554/eLife.41673.040

# Additional files
## Supplementary files
• Supplementary file 1. Metadata for the macrophage differentiation experiments performed in this study.
DOI: https://doi.org/10.7554/eLife.41673.023
• Supplementary file 2. Metadata for the RNA-seq samples generated in this study.
DOI: https://doi.org/10.7554/eLife.41673.024
• Transparent reporting form
DOI: https://doi.org/10.7554/eLife.41673.025

## Data availability
RNA-seq data from the acLDL stimulation study is available from ENA (PRJEB20734) and EGA (EGAS00001000876). RNA-seq data from the IFNɣ + Salmonella study is available from ENA (PRJEB18997) and EGA (EGAS00001002236). The imputed genotype data for HipSci cell lines is available from ENA (PRJEB11749) and EGA (EGAD00010000773). Processed data and QTL summary statistics are available from Zenodo: https://zenodo.org/communities/macrophage-tuqtls/.

The following datasets were generated:

| Author(s) | Year | Dataset title | Dataset URL | Database and Identifier |
|---|---|---|---|---|
| Alasoo K, Rodrigues J, Danesh J, | 2017 | Genetic effects on promoter usage are highly context-specific and | https://www.ebi.ac.uk/ena/data/view/ | European Nucleotide Archive, PRJEB20734 |

| Freitag DF, Paul DS, Gaffney DJ | | contribute to complex traits | PRJEB20734 | |
| Alasoo K, Rodrigues J, Danesh J, Freitag DF | 2017 | Genetic effects on promoter usage are highly context-specific and contribute to complex traits | https://www.ebi.ac.uk/ega/studies/EGAS00001000876 | European Genome-phenome Archive, EGAS00001000876 |

The following previously published datasets were used:

| Author(s) | Year | Dataset title | Dataset URL | Database and Identifier |
|---|---|---|---|---|
| Alasoo K, Rodrigues J, Mukhopadhyay S, Knights AJ, Mann AL, Kundu K, Hale C, Dougan G, Gaffney DJ | 2017 | Shared genetic effects on chromatin and gene expression indicate a role for enhancer priming in immune response | https://www.ebi.ac.uk/ena/data/view/PRJEB18997 | European Nucleotide Archive, PRJEB18997 |
| Alasoo K, Rodrigues J, Mukhopadhyay S, Knights AJ, Mann AL, Kundu K, Hale C, Dougan G, Gaffney DJ | 2017 | Shared genetic effects on chromatin and gene expression indicate a role for enhancer priming in immune response | https://www.ebi.ac.uk/ega/studies/EGAS00001002236 | European Genome-phenome Archive, EGAS00001002236 |
| Kilpinen H, Goncalves A | 2017 | Common genetic variation drives molecular heterogeneity in human iPSCs | https://www.ebi.ac.uk/ena/data/view/PRJEB11749 | European Nucleotide Archive, PRJEB11749 |
| Kilpinen H, Goncalves A | 2017 | Common genetic variation drives molecular heterogeneity in human iPSCs | https://www.ebi.ac.uk/ega/datasets/EGAD00010000773 | European Genome-phenome Archive, EGAD00010000773 |

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
