## [Decision Letter]

Thank you for submitting your article "Genetic effects on promoter usage are highly context-specific and contribute to complex traits" for consideration by *eLife*. Your article has been reviewed by three peer reviewers, and the evaluation has been overseen by Stephen Parker as the Reviewing Editor and Mark McCarthy as the Senior Editor. The following individual involved in the review of your submission has agreed to reveal their identity: Roger Pique-Regi (Reviewer #2).

The reviewers have discussed the reviews with one another and the Reviewing Editor has drafted this decision to help you prepare a revised submission.

Summary:

Most transcript usage QTL (tuQTL) studies do not differentiate between events leading to differential isoform usage, such as alternative promoter, alternative 3ʹ end, or alternative splicing that are a likely consequence of different molecular mechanisms. Alasoo and colleagues describe the development of such a tool – txrevise, which processes (Ensembl) transcript annotations to build an annotation of independent promoters, internal exons and 3' ends for use with an external tool such as Salmon for transcript quantification. They additionally map genetic associations for total gene expression, full length transcript usage and exon-exon junction using established methods and compare the approaches and results to those from txrevise. The authors show that promoter usage QTLs are generally more context specific and colocalize with genetic signals for complex traits. The approach is sound, and the methods/observations will be helpful to the field. The paper is well-written and we think will be appealing to a broad audience. However, there are several points that should be addressed, which we outline below.

Essential revisions:

1) There were two QTL scans done, where the initial scan happened over a smaller window. Was there any instance where the larger +/- 500 kb scan resulted in a stronger signal compared to the initial +/- 100 kb scan? If so, how were these treated?

2) More details about assignment of groups 1 and 2 during the txrevise process would be helpful. Surely there are examples that are not as simple as the one depicted in Figure 1—figure supplement 3. How are those more complicated cases treated? Another way of phrasing this: how do you decide when to choose more exons vs. fewer transcripts? Clarification of this approach will be helpful.

3) The word "group" may have different meanings in different sections of the txrevise methods. For example, in the last paragraph of the subsection “Quantifying transcriptional events with txrevise”, "group" describes the two different approaches to creating common scaffolds. At the end of the same section of the Materials and methods, it's not clear here what "group" means. Does it mean as above (the two different scaffold approaches) or one of the three different categories (promoter, internal, 3ʹ UTR)? Clarity here will be helpful for other labs that want to use this approach.

4) Not clear how multiple testing correction happens with txrevise – are you using the --grp-best flag across all the separate bits (promoter, internal, 3ʹ UTR) of a gene model, or only within the three different partitions? And what about across the two groups that are created? When mapping multiple QTLs, it is not clear if the authors map them all simultaneously or if they use a conditional on the lead QTL strategy. The authors should clarify this.

5) In Figure 2B, authors performed a replication analysis of QTLs. This appears to be only based on the LD between the lead QTL variants for different comparisons. However, the direction of effect size is ignored. This should also be included in all the replication analyses.

6) Given that promoter shifts are an important component of context specific tuQTL and are also enriched for complex traits, the authors could use their ATAC-seq data to further illustrate the mechanism and perhaps validate. Are changes on promoter usage dependent on the promoters being open? Do these types of tuQTL also have a QTL on ATAC-seq on the promoters? Even if the outcome is negative, which might indicate a more complex relationship between promoter usage and chromatin accessibility, this would further strengthen the manuscript.

7) The coupling part was uniformly perceived as weaker compared to the rest of the work and could be removed as it is a bit orthogonal to the main focus, especially considering it is not tied to any main figure.

---

## [Author Response]

Essential revisions:1) There were two QTL scans done, where the initial scan happened over a smaller window. Was there any instance where the larger +/- 500 kb scan resulted in a stronger signal compared to the initial +/- 100 kb scan? If so, how were these treated?

We have now modified the section “Association testing” in Materials and methods to clarify that only the lead variants identified in the initial +/- 100 kb scan were used for all downstream analyses. Since genes often have multiple independent eQTLs, we think it is likely that a larger scan in a +/- 500kb window will sometimes identify stronger signals. However, we did not quantify this, because the summary statistics from the larger window were only used for colocalisation analysis based on the lead variants identified in the smaller +/- 100kb. We did this to ensure that all of the variants in the +/- 200kb region around the lead QTL variant were included in the colocalisation analysis, because truncated summary statistics can lead to overestimation of the posterior probability of colocalisation.

2) More details about assignment of groups 1 and 2 during the txrevise process would be helpful. Surely there are examples that are not as simple as the one depicted in Figure 1—figure supplement 3. How are those more complicated cases treated? Another way of phrasing this: how do you decide when to choose more exons vs. fewer transcripts? Clarification of this approach will be helpful.

We agree that, in general, it is not easy to automatically decide whether to prioritise more shared exons between transcripts or larger number of transcripts; indeed, txrevise does not solve this particular problem. We have now modified the legend for Figure 1—figure supplement 3 to clarify that for each gene, txrevise always identifies two groups of transcripts and then performs all of the analyses across the two groups.

3) The word "group" may have different meanings in different sections of the txrevise methods. For example, in the last paragraph of the subsection “Quantifying transcriptional events with txrevise”, "group" describes the two different approaches to creating common scaffolds. At the end of the same section of the Materials and methods, it's not clear here what "group" means. Does it mean as above (the two different scaffold approaches) or one of the three different categories (promoter, internal, 3' UTR)? Clarity here will be helpful for other labs that want to use this approach.

We thank the reviewer for pointing out these inconsistencies. We have now edited the text in the Materials and methods section (subsection “Quantifying transcriptional events with txrevise”) and in the caption for Figure 1—figure supplement 3. In particular, the term “group” now always refers to the subset of transcripts used to identify the scaffold for event constructions. For the three different categories (promoter, internal exons, 3ʹ ends) we now consistently use the term “types of transcriptional events”.

4) Not clear how multiple testing correction happens with txrevise – are you using the --grp-best flag across all the separate bits (promoter, internal, 3' UTR) of a gene model, or only within the three different partitions? And what about across the two groups that are created? When mapping multiple QTLs, it is not clear if the authors map them all simultaneously or if they use a conditional on the lead QTL strategy.

We have now modified the “Association testing” subsection in the Materials and methods to clarify that the permutations were performed across the two subsets of transcripts used for event construction, thus reporting at most three tuQTLs per gene, i.e. one for each event type (promoters, internal exons, 3ʹ ends). After identifying permutation p-values with QTLtools, we also use Benjamini-Hochberg correction to account for the number of independent features that we have tested. Thus, even though we test 3x more features with txrevise, we account for that by using the Benjamini-Hochberg correction. We have also clarified that we mapped multiple QTLs only in the txrevise and Leafcutter analyses, because with those approaches we tested multiple event types (promoters, internal exons, 3ʹ ends) or multiple intron clusters per gene. We did not use conditional analysis.

5) In Figure 2B, authors performed a replication analysis of QTLs. This appears to be only based on the LD between the lead QTL variants for different comparisons. However, the direction of effect size is ignored. This should also be included in all the replication analyses.

The replication analysis is only presented to highlight the fact that different quantification approaches detect complementary sets of genetic associations. Including the effect size direction in our analysis can only reduce the number of QTLs that replicate between methods and not increase it. Therefore, this would not change the conclusions of our analysis.

Nonetheless, we agree that including effect direction in the replication analysis would be beneficial in principle. However, defining the direction of effect in such a way that is consistent between different quantification methods is challenging. For eQTLs, a reasonable definition would be the effect size of the alternative allele on the total read count originating from the gene. However, this becomes much more challenging for full-length transcript usage QTLs, because any variant that is associated with the increased usage of one transcript is by definition also associated with decreased usage of some other transcripts of the gene (since all transcripts have to sum up to 1). Thus, it is not clear which transcript should be used to define the direction of the effect. Similarly, it is challenging to compare the direction of effect sizes between Leafcutter tuQTLs and full-length tuQTLs, because it is not clear how to map Leafcutter splicing events to specific Ensembl transcripts, especially if Leafcutter analysis includes novel splice junctions not present in Ensembl. We now explicitly acknowledge this limitation of our analysis in the Materials and methods section (subsection “QTL replication between quantification methods”).

6) Given that promoter shifts are an important component of context specific tuQTL and are also enriched for complex traits, the authors could use their ATAC-seq data to further illustrate the mechanism and perhaps validate. Are changes on promoter usage dependent on the promoters being open? Do these types of tuQTL also have a QTL on ATAC-seq on the promoters? Even if the outcome is negative, which might indicate a more complex relationship between promoter usage and chromatin accessibility, this would further strengthen the manuscript.

We thank the reviewers for this excellent suggestion. We have now performed additional analysis assessing the overlap between promoter usage QTLs that we detect in our current study and chromatin accessibility QTLs that we have reported previously (Alasoo et al., 2018). These analyses revealed that 6% of the promoter usage QTLs are linked to putative coordinated changes in promoter accessibility (r^2^ > 0.9 between the lead variants) (subsection “Genomic properties of transcript usage QTLs”, third paragraph). To illustrate how this can yield mechanistic insight into promoter usage QTLs, we now also discuss one such example in more detail, as shown in Figure 2—figure supplement 7. We further describe this analysis in the Discussion, noting that although the overlap that we detected is small, this is likely influenced by low power in our chromatin accessibility dataset that contained only 41 individuals (Discussion, second paragraph).

7) The coupling part was uniformly perceived as weaker compared to the rest of the work and could be removed as it is a bit orthogonal to the main focus, especially considering it is not tied to any main figure.

We thank the reviewers for the suggestion. We initially added this section to justify the need for event-level analysis. However, we agree that it is orthogonal to the main focus of the paper and we have now removed this section as well as Supplementary Figure 8 from the manuscript. We have also removed the related section from the Discussion.